# Ageing and Osteoarthritis Synergically Affect Human Synoviocyte Cells: An In Vitro Study on Sex Differences

**DOI:** 10.3390/jcm11237125

**Published:** 2022-11-30

**Authors:** Francesca Veronesi, Deyanira Contartese, Veronica Borsari, Stefania Pagani, Milena Fini, Monica De Mattei, Matilde Tschon

**Affiliations:** 1Complex Structure Surgical Sciences and Technologies, IRCCS Istituto Ortopedico Rizzoli, 40136 Bologna, Italy; 2Scientific Direction, IRCCS Istituto Ortopedico Rizzoli, 40136 Bologna, Italy; 3Department of Medical Sciences, University of Ferrara, 44121 Ferrara, Italy

**Keywords:** fibroblast-like synoviocytes, osteoarthritis, aging, oxidative stress, sex

## Abstract

Osteoarthritis is a chronic inflammatory disease that affects all of the joints, especially those of the elderly. Aging is a natural and irreversible biological process implicated in the pathophysiology of many chronic diseases, such as osteoarthritis. Inflammation and oxidative stress are the main factors involved in osteoarthritis and aging, respectively, with the production of several pro-inflammatory cytokines such as Interleukin 1β (IL1β) and reactive oxygen species. The aim of the study was to set-up an in vitro model of osteoarthritis and aging, focusing on the sex differences by culturing male and female fibroblast-like synoviocytes (FLSs) with IL1β, hydrogen peroxide (H_2_O_2_), IL1β+H_2_O_2_ or a growth medium (control). IL1β+H_2_O_2_ reduced the cell viability and microwound healing potential, increased Caspase-3 expression and reactive oxygen species and IL6 production; IL1β increased IL6 production more than the other conditions did; H_2_O_2_ increased Caspase-3 expression and reactive oxygen species production; Klotho expression showed no differences among the treatments. The FLSs from female donors demonstrated a better response capacity in unfavorable conditions of inflammation and oxidative stress than those from the male donors did. This study developed culture conditions to mimic the aging and osteoarthritis microenvironment to evaluate the behavior of the FLSs which play a fundamental role in joint homeostasis, focusing on the sex-related aspects that are relevant in the osteoarthritis pathophysiology.

## 1. Introduction

Aging is an irreversible and inevitable biological process, which includes the deterioration of physical and mental conditions, and it is one of the main risk factors in the development of musculoskeletal aging-related diseases including osteoporosis (OP), intervertebral disc degeneration and osteoarthritis (OA) [1]. It is estimated that 10% of the world’s population who are over 60 years show significant clinical problems related to OA [2]. Among the different joint pathologies that affect the knee, OA is the most important chronic degenerative one, causing pain and disability, thus impacting their daily quality of life [3]. OA affects all of the joint components, such as the cartilage, subchondral bone, and synovial membrane, and it is characterized by a high production of pro-inflammatory cytokines, such as Interleukin 1β (IL1β). Aging is characterized by the accumulation of cellular senescence and of degenerative stimuli inside and outside the cell, while in OA that prevalently occurs with age, a progressive degeneration of the articular cartilage and surrounding joint tissues together with an inflamed microenvironment is observed [4].

A common feature of aging and OA is the imbalance between reactive oxygen species (ROS) production and the ability to repair tissue damages through endogenous antioxidant defenses. This causes an increase in the oxidative stress [5], which induces negative effects on the cells’ function (damages of proteins, nucleic acids and lipids) and decreases the tissue repair ability. In addition, senescent cells induce the adjacent cells to become so through a paracrine signaling pathway [6,7]. ROS are produced by the cartilage, synovium and subchondral bone cells, and they induce protein, DNA and lipid oxidative damage, promote chondrocyte senescence, cartilage destruction and inflammation and unbalance the subchondral bone remodeling [8]. In addition, ROS, produced by the mitochondria, directly affect the rate of telomere shortening, further accelerating cell senescence [9]. The two types of ROS which are implicated in aging and OA are superoxide (O^2−^) and hydrogen peroxide (H_2_O_2_) [10]. H_2_O_2_ at physiological concentrations enters into the cell metabolism to counteract the bacterial invasion, but at high concentrations, it produces other ROS that could lead principally to gene mutations, tumor formation and apoptosis [11,12].

Among the different proteins involved in aging and the OA process, Caspase-3 is one of the active executioners of apoptosis: its activation and intrinsic apoptotic pathways are up-regulated in an aged musculoskeletal system [13]. The higher production of cytokines and proteolytic enzymes, which increase the rate of cartilage degradation and synovial inflammation in OA pathology, increase Caspase-3 concentration in the synovial membrane, thereby becoming an OA biomarker [14].

Among the aging biomarkers, Klotho was identified for the first time in mice in 1997, and it was primarily expressed in reproductive organs, kidney and brain. Mice homozygous for the transgene have several phenotypes similar to the premature aging syndrome (calcification, pulmonary emphysema, osteoporosis, short life span and skin atrophy) [15,16]. It is an endogenous antagonist of the Wnt/β-catenin pathway which stimulates articular cartilage degradation, and consequently, its absence accelerates OA development [17]. The expression of Klotho is higher in normal mouse cartilage than it was in the OA model, in which Wnt/β-catenin and its target gene were up-regulated [18].

Interleukin-6 (IL6) is significantly higher in the synovial fluid and serum of patients affected by OA, and it locally increases the expression of metalloproteinases that degrade the cartilage and induce changes in the subchondral bone and synovial tissue inflammation [19]. In addition, older adults show high systemic IL6 levels [20].

In the joints of elderly or OA patients, the synovial membrane is altered, and this triggers the inflammatory pathway [21]. More precisely, in healthy conditions, the synovial membrane maintains the joint homeostasis because it with the fibroblast-like synoviocytes (FLSs) provides nourishment, protection and lubrication via hyaluronic acid production to the articular cartilage [22]. In OA and aging, the FLSs play a key role in joint degradation due to the production of pro-inflammatory cytokines and catabolic factors (such as proteases, and arachidonic acid metabolites), which contribute to cartilage damage, thereby increasing rate of the inflammatory and oxidative stress processes [23]. Despite the important role in the development of OA and joint aging, the FLSs, in comparison with chondrocytes, are even less studied [24].

In the literature until now, no study has evaluated the FLSs in the in vitro microenvironments of aging and OA.

Sex affects the pathophysiology and severity of several diseases, including OA [3]. The sex difference in OA is not well addressed, and it is still debated, especially from a cellular, molecular and mechanistic point of view due to the paucity of articles that have analyzed these aspects in both preclinical and clinical studies [25,26].

The hypothesis of our study was that OA, aging and sex show a more synergic effect on FLSs behavior in terms of viability, regenerative capacity and metabolic activity than that which has been observed for these conditions separately. For this purpose, in the present research study, we have fine-tuned an in vitro model of OA and aging by culturing FLSs obtained by patients affected by OA in selected microenvironments, thus potentially representing OA and aging through the exogenously added stimuli of IL1β and H_2_O_2_, alone or in combination. We decided to set up this preliminary in vitro model by purchasing commercially available primary cells from male and female OA patients in order to obtain similar certified baseline characteristics and limit the variability and ethical permission requirements. We used three cultures from male and female patients, and we acted on the culture microenvironment to reproduce OA or/and aging stimuli by adding IL1β and H_2_O_2_ alone or in combination. We selected a short experimental time to obtain a test system that would be able to be used as in vitro prescreening test for innovative drugs or medical devices before more complex in vitro and in vivo studies.

## 2. Materials and Methods

### 2.1. Cell Cultures

Human FLSs, isolated from male (M) and female (F) donors with knee OA, were purchased from Tebu-Bio (Cell Application, San Diego, CA, USA, n = 3 for each sex). The subjects were 50–67 years old for both of the sexes. The cells were cultured in synoviocyte growth medium (Cell Application) with 10% fetal bovine serum (FBS) (Euroclone, MI, Italy), 100 U/mL penicillin and 100 µg/mL streptomycin (Sigma Aldrich, St. Louis, MO, USA) in standard conditions (37 °C, 5% CO_2_/95% air, humidified atmosphere).

When they were confluent, the cells were detached using Trypsin/ethylendiaminetetracetic acid (EDTA) (Sigma Aldrich), counted in a Neubauer chambre using Erythrosin B vital dye (Alfa Aesar, Tewksbury, MA, USA), and seeded in 24-well plates at a density of 25,000 cells/cm^2^. Forty-eight hours after seeding, the growth medium was changed, and the following culture conditions were set up:-Control (CTR): the cells re-feed with synoviocyte growth medium;-IL1β: the cells cultured in synoviocyte growth medium were supplemented with 10 ng/mL IL1β (Cell Application) to replicate the inflammatory microenvironment present in OA conditions and to maintain the OA phenotype;-H_2_O_2_: the cells cultured in synoviocyte growth medium were supplemented with 30 µM H_2_O_2_ (Farmac-Zabban, Bologna, Italy) for the induction of the oxidative stress that is typical of aging;-IL1β+H_2_O_2_: the cells cultured in synoviocyte growth medium were supplemented with 10 ng/mL IL1β and 30 µM H_2_O_2_ to evaluate the combined effect of inflammatory microenvironment and oxidative stress.

### 2.2. Cell Viability Assay

An Alamar Blue assay (AbDSerotec, Oxford, UK) was used to evaluate the cell viability 24 h after the set-up of the conditions of the 4 groups. The reagent was added to each culture well (1:10 *v*/*v*) and incubated for 4 h at 37 °C. The fluorescence was read at 530ex-590em nm wavelengths using a Micro Plate reader (VICTOR X2030, Perkin Elmer, Milan, Italy) and expressed as relative fluorescence units (RFU).

### 2.3. In Vitro Microwound Healing Model

At the point of confluence, the FLSs cells were used in a monolayer for the preparation of an in vitro microwound healing model to analyze the ability of the cellular migration and regenerative capacities in disease-related conditions. In detail, 48 h after seeding, the cells were wounded by sliding a sterile 200-μm Eppendorf tip on the bottom of the well to create a cell free zone in the monolayer (at baseline, T0, wound dimension measured 654.3 ± 1 µm). After the cellular scratch was performed, the cultures were incubated with the different media as above reported, and they were observed using an inverted microscope (Nikon Eclipse Ti-U, Nikon Italia, Italy) equipped with a digital camera (Sight DS-Fi2, Nikon Italia, Italy) after 4 h (at T4) and 24 h (at T24). Each well was photographed at 4× magnification to cover the wounded width. The image acquisition software (NiSElements Advanced Research, Nikon Italia, Italy) was used to measure width of the cell free zone of the artificially created wound (10 measures for each image), and the percentage (%) of healing rate was calculated through the following formula [27]:$$healing\;percentage\;\left(\%\right)=\frac{T0-T24\;\mathrm{width}}{T0\;\mathrm{width}}*100$$

### 2.4. Gene Expression Analysis

After 24 h from the set-up of culture conditions, the total RNA was extracted using the RNeasy Mini Kit (Purelink™ RNA mini Kit, Ambion by Life Technologies, Carlsbad, CA, USA) according to manufacturer’s instructions. The total RNA was eluted with RNase-free water, quantified by NanoDrop 2000 (NANODROP 2720, Thermal Cycler, Applied Biosystem, Waltham, MA, USA), reverse transcribed to cDNA using Super Script VILO cDNA Synthesis kit (Invitrogen, by Thermo Fisher Scientific, Waltham, MA, USA) and diluted to the final concentration of 5 ng/μL. The gene expression was evaluated by semiquantitative polymerase chain reaction (PCR) analysis with SYBR green PCR master mix (QIAGEN GmbH, Hilden, Germany) in a Light Cycler 2.0 Instrument (Roche Diagnostics, GmbH, Manheim, Germany).

The protocol included: a denaturation cycle at 95 °C for 15 min, 25 to 40 cycles of amplification (95 °C for 15 s, an appropriate annealing temperature for each target gene for 20 s and 72 °C for 20 s) and a melting curve analysis to check for amplicon specificity. The Caspase-3 expression was evaluated as a marker of cellular apoptosis, while Klotho was evaluated as a biomarker of aging, and glyceraldehyde-3-phosphate dehydrogenase (GAPDH) was a reference gene. For this purpose, the following primers were used: Caspase-3 (CASP3) (QuantiTect Primer Assay Hs_CASP3_1_SG), α-Klotho (*α-KL*) (QuantiTect Primer Hs_KL_1_SG) and GAPDH (forward: 5′-TGGTATCGTGGAAGGACTCA-3′, reverse: 5′-GCAGGGATGATGTTCTGGA-3′). The mean threshold cycle was determined for each sample and used for the calculation of the relative expression using the Livak method (2^−ΔCt^), and we present the results in terms of the number of molecules of the gene of interest (GOI) per 100.000 molecules of the housekeeping gene GAPDH [28]. Each sample was tested in triplicate.

### 2.5. Immunoenzymatic Analysis

After 24 h, the cell cultures supernatants of all of the groups were collected and centrifuged to eliminate the particulates. Immunoenzymatic assays, an enzyme-linked immunosorbent assay (ELISA), were employed to quantify the amount of IL6 (pg/mL) (Fine Test) and ROS (IU/mL) (MyBioSource, San Diego, CA, USA) according to the manufacturer’s instructions.

### 2.6. Statistical Analysis

The data reported in the “Results” Section were obtained by normalizing the values of the IL1β, H_2_O_2_ and IL1β+H_2_O_2_ treatments for their respective CTR values. The values above 1 mean that the treatment has results that are above the CTR, and the values below 1 mean that the results are below the CTR.

All of the data are presented as the mean ± standard deviation (SD), and they refer to three independent experiments carried out in triplicate.

The statistical analysis of the data was conducted using SPSS v 29 software. After analyzing the homogeneity of variances, the one-way ANOVA test was used for multiple comparisons among the experimental groups by applying the post hoc tests by Tukey; the comparison between the males and females within each experimental group (IL1β, H_2_O_2_ and IL1β+H_2_O_2_) was carried out using the Student’s t test for unpaired data; the significance level was set at *p* ˂ 0.05.

## 3. Results

### 3.1. Cell Viability

At 24 h (T24), the Alamar Blue assay showed that IL1β+H_2_O_2_ treatment significantly reduced the cell viability in comparison to IL1β and H_2_O_2_ in both the male (*, *p* < 0.05) and female cells (***, *p* < 0.0005). The female cells showed a significantly higher viability than the male ones did in the IL1β treatment (a, *p* < 0.05) (Figure 1).

### 3.2. Cell Migration

At T24, after the cellular scratch, it was observed that the IL1β+H_2_O_2_ treatment significantly decreased the rate of cellular migration, and therefore, this decreased the percentage of microwound healing as compared to IL1β and H_2_O_2_ treatments in the both male (***, *p* < 0.0005) and female (**, *p* < 0.005) cells. The female cells showed a higher migration rate than the male ones did in the IL1β+H_2_O_2_ (a, *p* < 0.05) treatment (Figure 2 and Figure 3).

### 3.3. Gene Expression

After 24 h, the Caspase-3 expression, which was evaluated by semiquantitative PCR analysis, was significantly higher in the male cells in the H_2_O_2_ and IL1β+H_2_O_2_ treatments than in the IL1β one (***, *p* < 0.0005). In the female cells, the H_2_O_2_ treatment showed higher Caspase-3 levels than they did in the IL1β and IL1β+H_2_O_2_ treatments (***, *p* < 0.0005) (Figure 4a). No significant differences were observed between the sexes for all of the treatments (Figure 4a).

The Klotho expression, which was evaluated by semiquantitative PCR analysis, showed no difference among the treatments within each sex (Figure 4b). The female cells showed a significantly higher Klotho expression than the male ones did for all of the treatments (Figure 4b).

### 3.4. Immunoenzymatic Test

At T24, the immunoenzymatic assays showed that, for the male cells, the IL1β and IL1β+H_2_O_2_ treatments significantly increased the IL6 production more than the H_2_O_2_ treatment did (***, *p* < 0.0005) (Figure 5a). For the female cells, the IL1β treatment showed a higher IL6 production than the H_2_O_2_ (***, *p* < 0.0005) and IL1β+H_2_O_2_ treatments did (**, *p* < 0.005). Additionally, the IL1β+H_2_O_2_ treatment induced higher IL6 production than the H_2_O_2_ treatment did (**, *p* < 0.005) (Figure 5a). The female cells showed higher IL6 levels than the males ones in the IL1β treatment (a, *p* < 0.05) (Figure 5a).

In regard to the ROS, in the male cells, the H_2_O_2_ and IL1β+H_2_O_2_ treatments significantly increased the ROS production more than the IL1β treatment did (***, *p* < 0.0005) (Figure 5b). In the female cells, the H_2_O_2_ (*, *p* < 0.05) and IL1β+H_2_O_2_ (***, *p* < 0.0005) treatments showed significantly higher ROS levels than the IL1β treatment did (Figure 5b). In the H_2_O_2_ and IL1β+H_2_O_2_ treatments, the male cells showed significantly higher ROS levels than the female ones did (a, *p* < 0.05) (Figure 5b).

## 4. Discussion

Aging is considered to be a progressive decline of the physiological functions with a concomitant increase of disease development, such as cancer, OA, dementia, atherosclerosis, infection, or cell death. Age is the primary risk factor for OA: the continuous use of the joints over the entire lifespan of the individual causes their progressive deterioration that, coupled with the poor regenerative healing capacities of the articular cartilage, contributes to the development and progression of OA [29,30].

The results of the present study show that: (1) the association of IL1β and H_2_O_2_ (IL1β+H_2_O_2_) significantly reduced the cell viability and microwound healing percentage in both of the sexes; (2) the Caspase-3 expression and ROS production were higher after the H_2_O_2_ and H_2_O_2_+IL1β treatments; (3) IL1β and IL1β +H_2_O_2_ determined an increase in the IL6 production in both of the sexes; (4) the Klotho expression showed no difference between the treatment groups in either of the sexes; (5) the female cells showed significantly higher cell viability and IL6 production in the IL1β treatment, higher Klotho in all of the treatments, and lower ROS production in the H_2_O_2_ and IL1β+H_2_O_2_ treatment groups. In addition, also the microwound healing percentage was higher in the female cells than it was in male cells in the IL1β+H_2_O_2_ treatment.

Until now, only a limited number of studies have evaluated the behavior of knee FLSs in an oxidative stress situation, but they have regarded FLSs as being isolated from rheumatoid arthritis (RA) patients and exposed to different H_2_O_2_ concentrations and timings for the modelling of the RA microenvironment in vitro. Similarly to the present study, some of the authors have showed a reduced cell viability and a decrease in some of the proteins that protect the cells from oxidative stress and an increase in Caspase-3, apoptosis and mitochondrial ROS [31,32,33,34,35,36,37]. In the literature, many different concentrations of H_2_O_2_ have been used for different experimental times. We chose a 30 µM concentration of H_2_O_2_ thanks to our preliminary assays which we performed to identify the best concentration of oxidative stress stimulus without causing cell death. In our previous studies, co-culture models of osteoblasts and osteoclasts treated with H_2_O_2_ were performed to simulate oxidative stress, and consequently, cellular senescence and OP [38,39]. Our results showed that H_2_O_2_ alone increased Caspase-3 expression and ROS production, also, in association with IL1β. H_2_O_2_ alone did not cause cell death, but when it was associated with IL1β, it reduced cell viability and microwound healing percentage.

On the other hand, H_2_O_2_ did not increase the IL6 levels, but when it was associated with IL1β, it increased the production of IL6. In support of this result, there are an extensive studies in the literature which claim that IL1β, present in a joint affected by OA, induces the production of IL6, one of the most important biomarkers present in the synovial fluid [40,41].

In the present in vitro study, we added IL1β at a concentration of 10 ng/mL to reproduce an OA microenvironment, which is a well-established in vitro model in the literature [42], and has already been performed in our previous in vitro studies [42,43]: in these experimental settings, the cells from either healthy or OA donors behaved in the same way, pointing to the strength of our experimental culture models and their reliability in reproducing an OA environment, independent of the healthy or OA origin of the joint cells. As suggested by Blasioli and Kaplan, in, in vitro culture systems, this concentration of IL1β is needed to reproduce an environment that is like OA [44]. Yet, he addition of H_2_O_2_ to the cell cultures is employed as an in vitro model of cell senescence to study the effect of oxidative stress or stress-induced senescence on several different cell types. Focusing on knee joint cells, H_2_O_2_ is added in cultures of human chondrocytes, harvested from healthy or OA patients and the chondrocyte cell line [45,46,47,48,49,50].

In comparison with CTR, the treatment with IL1β alone determined an increased IL6 production, while H_2_O_2_ alone caused an increase in Caspase-3 expression and ROS production. The combined treatment with IL1β and H_2_O_2_ induced lower cell viability, higher Caspase-3 expression and IL6 and ROS production than CTR did.

Regarding the sex differences, in our recent reviews, the clinical studies revealed that females affected by OA showed a lower knee size and curvature, cartilage volume and thickness, and higher alterations in the joint functionality, fall risk, visual analogue scale (VAS) score, pro-inflammatory cytokines and drugs taken in comparison with the males [25]. The sex differences of FLSs in presence of senescence and OA conditions have not been yet investigated in the literature. Our recently published literature review [27] collected in vitro and in vivo studies on OA found a lack of consistency and evidence-based preclinical results which could help in understanding the role of sex in the etiology, onset, molecular events and signaling pathways of OA. The only one in vitro study that used FLSs from female rats, which were affected by temporomandibular OA and cultured in presence of tumor necrosis factor α (TNFα), showed higher inducible nitric oxide synthase (iNOS), IL1β and monocyte chemoattractant protein-1 (MCP1) expression compared to the cells of the male rats in the same conditions [51]. Differently from that study involving rat derived FLSs, we found that the fibroblasts from female subjects showed a higher cell viability, microwound healing and Klotho expression and lower ROS production than those from the male subjects. On the other hand, the IL6 production was significantly higher in the female cells than it was in the male ones. However, this topic is still much debated in the literature, indicating the necessity to perform more preclinical and clinical studies to elucidate the sex-based determinants and mechanisms in view of developing sex specific protocols and tailored drugs [26].

Some recent preclinical studies in the literature evaluated other age-related diseases of the musculoskeletal system, such as OP. One study observed that proinflammatory and senescent subtypes of immune cells were higher in the bone marrow of aged rats and mice, and that grancalcin induced senescence and an imbalance in osteogenesis versus adipogenesis [52]. In the context of OP, engineered exosomes, BT-Exo-siShn3, showed an anti-OP effect because they enhanced osteogenic differentiation and decreased autologous RANKL expression, thereby inhibiting osteoclast formation [53]. In addition, bioinspired nanovesicles from human induced pluripotent stem cell-derived endothelial cells were engineered to secret cytokines, favoring osteogenesis and anti-inflammation. It induced the osteogenic differentiation of the bone mesenchymal stem cells (BMSCs) and the M1-macrophage-dominant pro-inflammatory microenvironment in the osteoporotic bones in ovariectomized mice [54].

Finally, a recent review of the literature elucidated the role of subchondral bone in OA, showing that the abnormal subchondral bone remodeling, angiogenesis and sensory nerve innervation contribute directly or indirectly to cartilage destruction and pain in OA [55].

One of the strengths of the present in vitro study is that it is compliant with the 3R (reduction, refinement and replacement) principles for animal use for scientific purposes, which were first defined by Russel and Burch [56], because it is performed in an advanced in vitro model, instead or before of in vivo one. Generally, an in vitro environment provides information about cell–cell and cell–extracellular matrix interactions, as well as cell surface receptors expression, growth factor synthesis and the physical and chemical conditions of the healthy and diseased subjects. By increasing the in vitro complexity, several experimental studies highlighted the capability of in vitro culture systems to represent and recapitulate the characteristics of many pathologies [57,58,59], and to preliminary gather, not only data about the safety and cytocompatibility, but also data regarding the bioactivity and therapeutic efficacy of the newly developed materials, medical devices or therapies.

Secondly, we standardized the present in vitro model, starting with the commercial cell lines retrieved from OA patients with the same age range to limit the variability as much as possible between the patients (inter-variability), avoiding ethical implications, i.e., the need to isolate a huge number of cells from many patients with different ages and degrees of OA.

However, our study also has limitations. Firstly, the number of samples was low, and therefore, it is not representative of the entire population, and the selected experimental times were short. In addition, this in vitro model was built in a two-dimensional culture well, starting from a limited number of patients and with an oversimplified set-up that was greatly different from the complex in vivo situation, in which there are many cellular phenotypes and many hormonal, chemical, biological and mechanical stimuli. However, in the author’s opinion, this model could help in the first instance to evaluate, in a controlled manner, the contribution of an OA and an aging simplified microenvironment, taking into consideration the sex differences without using in vivo models. Subsequent in vivo studies will be set up to verify the in vitro results proposed by the present study, that will allow for the analysis of a greater number of data, making it possible to clarify the complex mechanisms of this pathology associated with the age and sex differences.

## 5. Conclusions

To conclude, in summary, this study has developed culture conditions to mimic aging and the OA microenvironment found in the knee joint to evaluate the behavior of FLSs, which are cells that play a fundamental role in the homeostasis of the joint. Relevance was also attributed to the sex differences in this in vitro model by testing both male and female FLSs from OA patients. It was observed that the combined effect of the inflammatory microenvironment and oxidative stress (IL1β+H_2_O_2_) reduced the cell viability and microwound healing percentage and increased the Caspase-3 expression and ROS and IL6 production; the inflammatory microenvironment present in the OA conditions (addition of IL1β alone) increased the IL6 production more than the other conditions did; the oxidative stress that is typical of aging (H_2_O_2_) increased Caspase-3 expression and ROS production; the Klotho expression showed no differences among the treatments.

This study also evaluated sex differences indicating that the FLSs from female donors demonstrated a better response capacity in unfavorable conditions of inflammation and oxidative stress than those from male donors did. The female cells showed higher cell viability, microwound healing percentage and Klotho expression and lower ROS production than the male cells did.

## Figures and Tables

**Figure 1 jcm-11-07125-f001:**
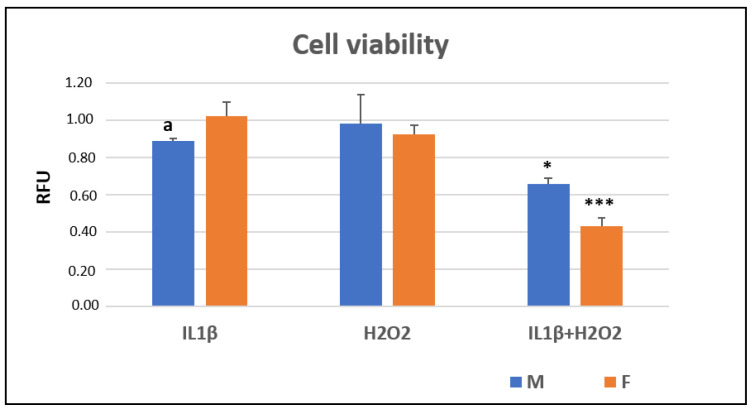
Cell viability (relative fluorescence units—RFU) in IL1β, H_2_O_2_ and IL1β+H_2_O_2_ treatments normalized for CTR in male (M) and female (F) FLSs after 24 h (mean ± standard deviation, n = 3). One-way ANOVA followed by Tukey post hoc test: ***, *p* < 0.0005 among treatments in F: IL1β+H_2_O_2_ vs. IL1β/H_2_O_2_. *, *p* < 0.05 among treatments in M: IL1β+H_2_O_2_ vs. IL1β/H_2_O_2_. Student’ *t*-test: a, *p* < 0.05: M vs. F for IL1β+H_2_O_2_.

**Figure 2 jcm-11-07125-f002:**
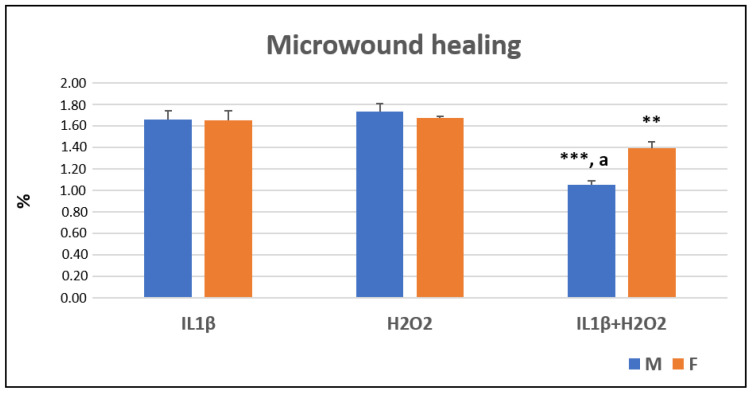
Microwound healing percentages (%) in IL1β, H_2_O_2_ and IL1β+H_2_O_2_ treatments normalized for CTR in male (M) and female (F) FLSs at T24 (mean ± standard deviation, n = 3). One-way ANOVA followed by Tukey post hoc test: ***, *p* < 0.0005 among treatments in M: IL1β+H_2_O_2_ vs. IL1β/H_2_O_2_. **, *p* < 0.005 among treatments in F: IL1β+H_2_O_2_ vs. IL1β/H_2_O_2_. Student’ *t*-test: a, *p* < 0.05: M vs. F for IL1β+H_2_O_2_.

**Figure 3 jcm-11-07125-f003:**
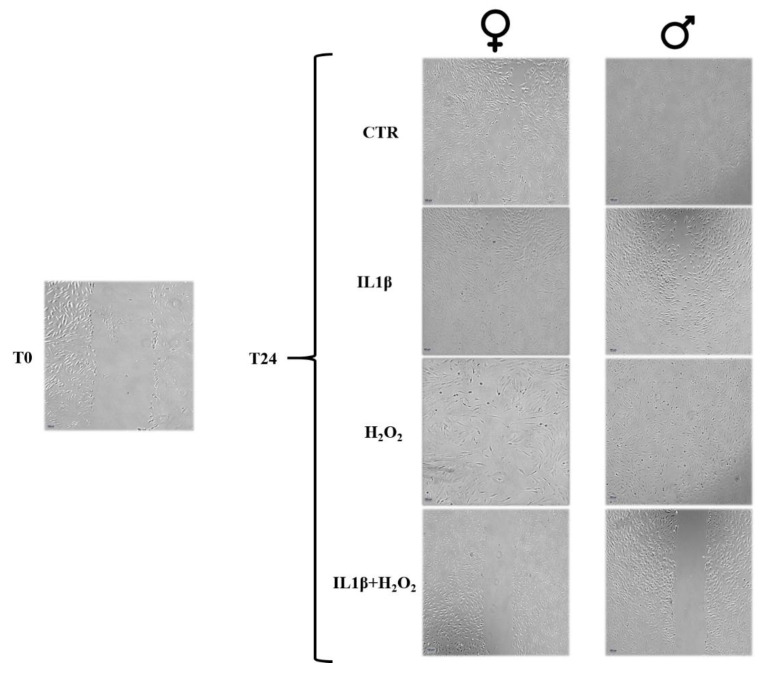
Microscopic images of the microwounds at T0 and T24 for IL1β, H_2_O_2_ and IL1β+H_2_O_2_ treatments in male and female FLSs. Magnification 4×. At T24, images show a trend of cellular growth into the microwound (healing potential) lower in IL1β+H_2_O_2_ treatment than in IL1β and H_2_O_2_ treatments in both male and female cells. Scale bar = 100 µm.

**Figure 4 jcm-11-07125-f004:**
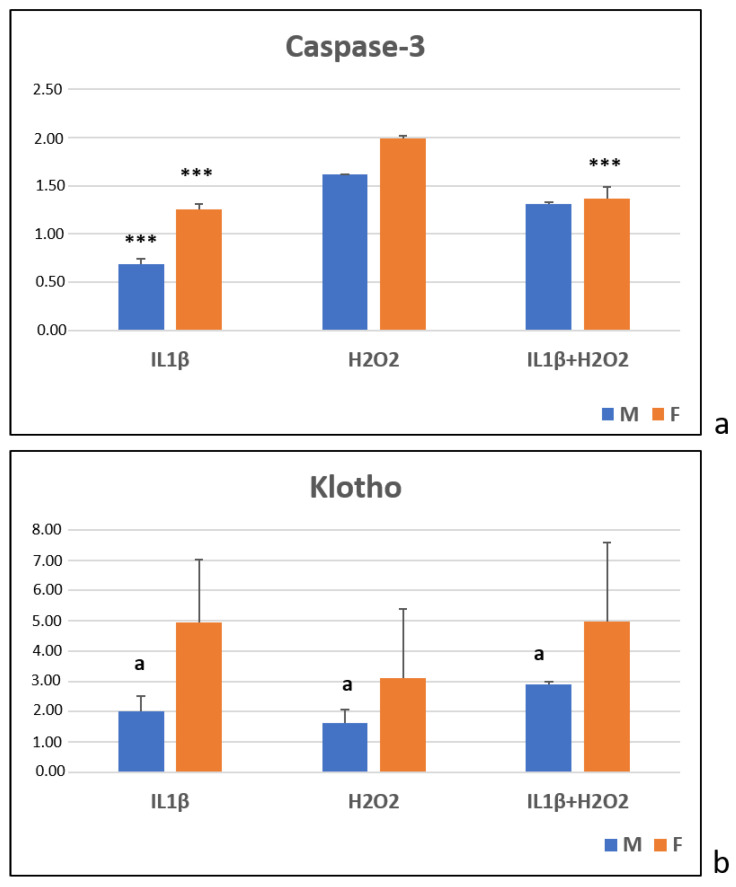
Caspase-3 (**a**) and Klotho (**b**) gene expression in IL1β, H_2_O_2_ and IL1β+H_2_O_2_ treatments normalized for CTR in male (M) and female (F) FLSs at T24 (mean ± standard deviation, n = 3). (**a**) One-way ANOVA followed by Tukey post hoc test: (**a**) ***, *p* < 0.0005 among treatments in M: IL1β vs. H_2_O_2_ and IL1β+H_2_O_2_. ***, *p* < 0.0005 among treatments in F: IL1β and IL1β+H_2_O_2_ vs. H_2_O_2_. (**b**) Student’ *t*-test: a, *p* < 0.05: M vs. F for all treatments.

**Figure 5 jcm-11-07125-f005:**
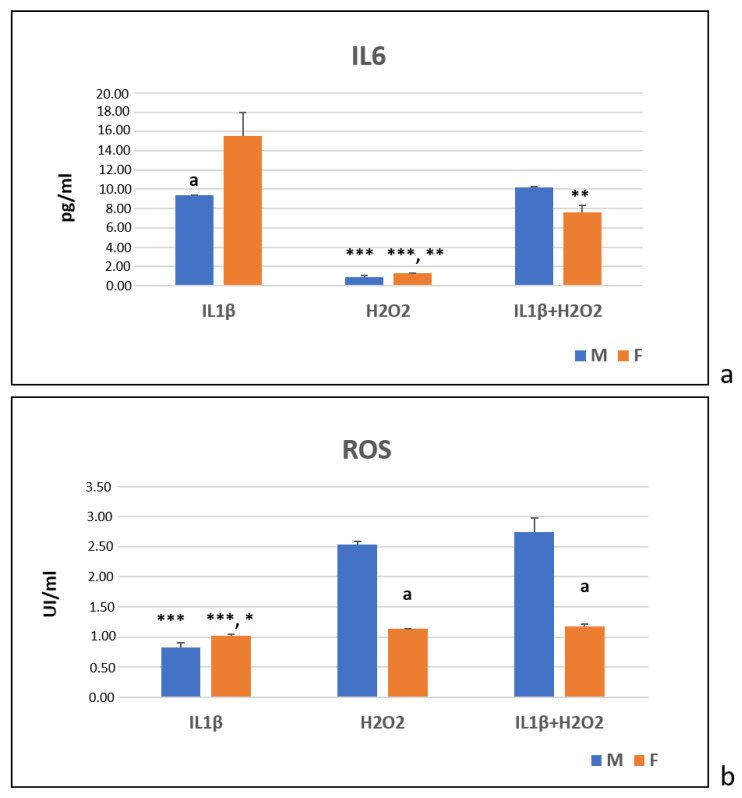
IL6 (pg/mL) (**a**) and ROS (UI/mL) (**b**) protein production in IL1β, H_2_O_2_ and IL1β+H_2_O_2_ treatments normalized for CTR in male (M) and female (F) FLSs at T24 (mean ± standard deviation, n = 3). (**a**) One-way ANOVA followed by Tukey post hoc test: ***, *p* < 0.0005 among treatments in M: H_2_O_2_ vs. IL1β and IL1β+H_2_O_2_. ***, *p* < 0.0005 among treatments in F: H_2_O_2_ vs. IL1β; **, *p* < 0.005: H_2_O_2_ vs. IL1β+H_2_O_2_; **, *p* < 0.005: IL1β+H_2_O_2_ vs. IL1β. Student’ *t*-test: a, *p* < 0.05: M vs. F for IL1β. (**b**) One-way ANOVA followed by Tukey post hoc test: ***, *p* < 0.0005 among treatments in M: IL1β vs. IL1β+H_2_O_2_ and H_2_O_2_. *, *p* < 0.05 among treatments in F: IL1β vs. H_2_O_2_; ***, *p* < 0.0005: IL1β vs. IL1β+H_2_O_2_. Student’ *t*-test: a, *p* < 0.05: F vs. M for H_2_O_2_ and IL1β+H_2_O_2_.

## Data Availability

The data presented in this study are available on request from the corresponding author.

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
