# Peer review of "Ageing and Osteoarthritis Synergically Affect Human Synoviocyte Cells: An In Vitro Study on Sex Differences"

_jcm, 2022, doi:10.3390/jcm11237125_

Round 1

Reviewer 1 Report

I'm very glad to become acquainted with such interesting research. Hope that my review will help to improve the quality of the article. Despite major efforts, there are notes.

Where CTR group on Figures?

Doses of IL1β and H2O2 should be explained. Why 10 ng/ml and 30 µM?

Abbreviations OA and ROS should be deleted form Abstract.

Figures should be improved.

It's hard to judge Discussion without fully presented data.

Author Response

I'm very glad to become acquainted with such interesting research. Hope that my review will help to improve the quality of the article. Despite major efforts, there are notes.

Where CTR group on Figures?

All data reported on the different treatments in figures were normalized for their respective CTR values, as indicated in the “Statistical analysis” paragraph: “Data reported in the “Results” section were obtained by normalizing values of the IL1β, H2O2 and IL1β+H2O2 treatments for their respective CTR values.”. Therefore, the CTR group is not shown in figures as histogram. Values above 1 mean that the treatment has results above the CTR and values below 1 mean results below the CTR. We have decided to normalize the results of all the treatment groups for the CTR because each experimental group had its CTR value different from the others; so far, to be able to compare different groups of treatment and male cells with female cells, we decided to normalize their values.

So, the following sentence has been added to the “Statistical analysis” paragraph: “Values above 1 mean that the treatment has results above the CTR and values below 1 mean results below the CTR” (page 4, lines 178-180).

However, as suggested by the reviewer, we improved the “Discussion” section with a comment about the CTR and the trend of the treatment values respect to CTR for each analysed parameter. This is included in the reviewer's last question.

Doses of IL1β and H2O2 should be explained. Why 10 ng/ml and 30 µM?

The H2O2 concentration was chosen based on preliminary experiments we performed to identify the most effective oxidative stress stimulus without causing cell death. The concentration of IL1β was chosen after careful research in the literature to identify the most effective quantity used to reproduce an inflammatory microenvironment typical of OA. In addition, in our previous in vitro studies, we have yet employed this IL1β concentration to reproduce an inflammatory microenvironment typically seen in a joint affected by OA.

So, some sentences have been added to the “Discussion” section as follows: “In literature, many different concentrations of H2O2 have been used for different experimental times. We chose 30 µM concentration of H2O2 thanks to our preliminary assays performed to identify the better concentration of oxidative stress stimulus without causing cell death. In our previous studies, co-culture models of osteoblasts and osteoclasts treated with H2O2 were performed to simulate oxidative stress and consequent cellular senescence and OP [38,39].” (page 9, lines 292-297).

“In the present in vitro study, we added IL1β, at a concentration of 10 ng/ml, to reproduce an OA microenvironment, that is a well-established in vitro model in literature [42], already performedin our previous in vitro studies [42, 43]: in these experimental settings, cells from either healthy or OA donors behaved in the same way, pointing to the strength of our experimental culture models and its reliability in reproducing an OA environment, independent of the healthy or OA origin of the joint cells. As suggested by Blasioli and Kaplan, in in vitro culture systems this concentration of IL1β is needed to reproduce an environment like OA [44].” (page 9, lines 305-312).

In addition, a new reference has been added to the “References” section as follows:

“44.     Blasioli, D.J.; Kaplan, D.L. The roles of catabolic factors in the development of osteoarthritis. Tissue Eng Part B Rev 2014, 20, 355–363.” (page 12, lines 498-499).

In addition, the reference number 53 became number 43.

Abbreviations OA and ROS should be deleted form Abstract.

As suggested by the reviewer, we deleted the abbreviations OA and ROS from Abstract.

Figures should be improved.

As suggested by the reviewer, we improved the figures quality.

It's hard to judge Discussion without fully presented data.

As suggested by the reviewer, we modified the “Discussion” section as follows: “In comparison with un-treated control (CTR), the treatment with IL1β alone deter-mined increased IL6 production, while H2O2 alone caused an increase in Caspase-3 ex-pression and ROS production. The combined treatment with IL1β and H2O2 induced low-er cell viability, higher Caspase-3 expression and IL6 and ROS production than CTR.” (page 9, lines 317-320).

Reviewer 2 Report

This study developed culture conditions to mimic aging and OA microenvironment, to evaluate the behavior of FLSs, that play a fundamental role in joint homeostasis and focusing on sex-related aspects that are relevant in the OA pathophysiology. However, the amount of data in the article is very small. There is no animal experiment or mechanism discussion, and only one phenomenon is reported.

The title is not accurate enough. It is recommended to modify the title

Author Response

This study developed culture conditions to mimic aging and OA microenvironment, to evaluate the behavior of FLSs, that play a fundamental role in joint homeostasis and focusing on sex-related aspects that are relevant in the OA pathophysiology. However, the amount of data in the article is very small. There is no animal experiment or mechanism discussion, and only one phenomenon is reported.

As observed by the reviewer, there is no animal experiment. However, the aim of the present study was to observe gender differences in cells (synoviocytes), in the development of knee joint OA and aging, without resorting to animal study. Although these cells play an important role in the joint, they have not yet been studied extensively in the literature. This is a preliminary study, and we are currently working on osteoarthritic and aged animals, but the results will be published in the near future. So, we now have included these issues as study limitations in the “Discussion” section as follows: “Subsequent in vivo studies will be set up to verify the in vitro results proposed by the present study, that will allow for the analysis of a greater number of data, making it possible to clarify the complex mechanisms of this pathology associated with age and sex differences.” (page 10, lines 363-366).

The title is not accurate enough. It is recommended to modify the title.

As suggested by the reviewer, we have modified the title of manuscript as follows: “Ageing and osteoarthritis synergically affect human synoviocyte cells: an in vitro study on gender differences”.

Round 2

Reviewer 1 Report

The article was revised as required.

Author Response

We want to thank the reviewer for her/his suggestions

Reviewer 2 Report

The manuscript titled “Ageing and osteoarthritis synergically affect human synoviocyte cells: an in vitro study on gender differences” is comparatively detailed and presents an interesting technique to build an in vitro model of aging-related osteoarthritis. The results are not abundant but explained clearly, and the conclusions are drawn based on the provided results only. This study was comparatively novel and would also be a good fit for Journal of Clinical Medicine. I have only a few minor comments regarding the manuscript.

1.The researchers should carefully check their manuscript for errors and omissions, including the full names of some acronyms when they were mentioned firstly in the manuscript.

2. Figure 3 should supplement the measuring scale of cell migration.

3. To present a more insightful discussion part, the authors should compare the in-vitro studies of other bone metabolic and age-related diseases such as osteoporosis.The authors are suggested to refer to these works:Cell Metab 2021, 33 (10), 1957-1973;Bioactive Materials 10 (2022) 207–221; ACS Nano 2022, 16, 11076−11091;Bone Res 2021, 9 (1), 20.

Author Response

The manuscript titled “Ageing and osteoarthritis synergically affect human synoviocyte cells: an in vitro study on gender differences” is comparatively detailed and presents an interesting technique to build an in vitro model of aging-related osteoarthritis. The results are not abundant but explained clearly, and the conclusions are drawn based on the provided results only. This study was comparatively novel and would also be a good fit for Journal of Clinical Medicine. I have only a few minor comments regarding the manuscript.

1.The researchers should carefully check their manuscript for errors and omissions, including the full names of some acronyms when they were mentioned firstly in the manuscript.

As rightly suggested by the reviewer, the manuscript has been now checked for errors and omissions. The corrections are in red in the text.

2. Figure 3 should supplement the measuring scale of cell migration.

Now Figure 3 has been modified.

3. To present a more insightful discussion part, the authors should compare the in-vitro studies of other bone metabolic and age-related diseases such as osteoporosis. The authors are suggested to refer to these works:Cell Metab 2021, 33 (10), 1957-1973;Bioactive Materials 10 (2022) 207–221; ACS Nano 2022, 16, 11076−11091;Bone Res 2021, 9 (1), 20.

In accordance with the reviewer's advice, the “Discussion” section has been improved as follows: “Some recent literature preclinical studies evaluated other age-related diseases of musculoskeletal system, such as OP. One study observed that proinflammatory and senescent subtypes of immune cells were higher in the bone marrow of aged rats and mice, and that grancalcin induced senescence and an imbalance in osteogenesis versus adipogenesis [52]. In the context of OP, engineered exosomes BT-Exo-siShn3 showed an anti-OP effect because it enhanced osteogenic differentiation, decreased autologous RANKL ex-pression, inhibiting osteoclast formation [53]. In addition, bioinspired nanovesicles from human induced pluripotent stem cells-derived endothelial cells were engineered to secret cytokines favoring osteogenesis and anti-inflammation. It induced the osteogenic differentiation of bone mesenchymal stem cells (BMSCs) and the M1-macrophage-dominant pro-inflammatory microenvironment in osteoporotic bones in ovariectomized mice [54]. Finally, a recent review of the literature elucidated the role of subchondral bone in OA, showing that the abnormal subchondral bone remodeling, angiogenesis and sensory nerve innervation contribute directly or indirectly to cartilage destruction and pain in OA [55].” (page 10, lines 341-354).

In addition, the following references have been added to the “References” section as follows: “

  1. Li, C.J.; Xiao, Y.; Sun, Y.C.; He, W.Z.; Liu, L.; Huang, M.; He, C.; Huang, M.; Chen, K.X.; Hou, J.; Feng, X.; Su, T.; Guo, Q.; Huang, Y.; Peng, H.; Yang, M.; Liu, G.H.; Luo, X.H. Senescent immune cells release grancalcin to promote skeletal aging. Cell Metab 2021, 33, 1957-1973.
  2. Cui, Y.; Guo, Y.; Kong, L.; Shi, J.; Liu, P.; Li, R.; Geng, Y.; Gao, W.; Zhang, Z.; Fu, D. A bone-targeted engineered exosome platform delivering siRNA to treat osteoporosis. Bioact Mater 2021, 10, 207-221.
  3. Cui, Y.; Li, Z.; Guo, Y.; Qi, X.; Yang, Y.; Jia, X.; Li, R.; Shi, J.; Gao, W.; Ren, Z.; Liu, G.; Ye, Q.; Zhang, Z.; Fu, D. Bioinspired Nanovesicles Convert the Skeletal Endothelium-Associated Secretory Phenotype to Treat Osteoporosis. ACS Nano 2022,16, 11076-11091.
  4. Hu, Y.; Chen, X.; Wang, S.; Jing, Y.; Su, J. Subchondral bone microenvironment in osteoarthritis and pain. Bone Res 2021, 9, 20.” (page 13, lines 532-541).

Accordingly, the other references have been renumbered.